Detection of stable community structures within gut microbiota co-occurrence networks from different human populations

Jackson Matthew A. 1
Bonder Marc Jan 2
Kuncheva Zhana 3
Zierer Jonas 1 4
Fu Jingyuan 2 5
Kurilshikov Alexander 2
Wijmenga Cisca 2 6
Zhernakova Alexandra 2
Bell Jordana T. 1
Spector Tim D. 1
Steves Claire J. claire.j.steves@kcl.ac.uk 1
1 Department of Twin Research & Genetic Epidemiology, King’s College London , London , United Kingdom
2 University Medical Center Groningen, Department of Genetics, University of Groningen , Groningen , Netherlands
3 Department of Mathematics, Imperial College London , London , United Kingdom
4 Institute of Bioinformatics and Systems Biology, Helmholtz Zentrum München , Neuherberg , Germany
5 University Medical Center Groningen, Department of Pediatrics, University of Groningen , Groningen , Netherlands
6 K.G. Jebsen Coeliac Disease Research Centre, Department of Immunology, University of Oslo , Oslo , Norway
Ettrich Rüdiger
Electronic publication date: 2018 Feb 7
Publication date: 2018
Volume: 6
Electronic Location ID: e4303
Received 2017 Oct 23; Accepted 2018 Jan 10
Copyright: ©2018 Jackson et al.
Copyright year: 2018
Copyright holder: Jackson et al.
License: This is an open access article distributed under the terms of the Creative Commons Attribution License, which permits unrestricted use, distribution, reproduction and adaptation in any medium and for any purpose provided that it is properly attributed. For attribution, the original author(s), title, publication source (PeerJ) and either DOI or URL of the article must be cited.
License URL: https://creativecommons.org/licenses/by/4.0/

Keywords: Microbiome, Gut, Human, Population, Networks, Co-occurrence, Communities

Funding: National Institutes of Health (NIH) RO1 DK093595 DP2 OD007444 Wellcome Trust; European Community’s Seventh Framework Programme (FP7/2007-2013) National Institute for Health Research (NIHR)-funded BioResource Clinical Research Facility and Biomedical Research Centre Chronic Disease Research Foundation (CDRF) Engineering and Physical Sciences Research Council (EPSRC) I 02/19 Top Institute Food and Nutrition, Wageningen TiFN GH001 Netherlands Organization for Scientific Research (NWO) NWO-VIDI 864.13.013 CardioVa sculair Onderzoek Nederland ERC-START grant ERC-715772 Rosalind Franklin Fellowship The TwinsUK microbiota project was funded by the National Institutes of Health (NIH) RO1 DK093595, DP2 OD007444. TwinsUK received funding from the Wellcome Trust; European Community’s Seventh Framework Programme (FP7/ 2007-2013), the National Institute for Health Research (NIHR)-funded BioResource, Clinical Research Facility and Biomedical Research Centre based at Guy’s and St Thomas’ NHS Foundation Trust in partnership with King’s College London. Claire J. Steves is funded under a grant from the Chronic Disease Research Foundation (CDRF). Zhana Kuncheva was partially supported by the Engineering and Physical Sciences Research Council (EPSRC) and by grant no. I 02/19 of Bulgarian NSF. The Lifelines-DEEP project was funded by grants from the Top Institute Food and Nutrition, Wageningen, the Netherlands, to Cisca Wijmenga (TiFN GH001); the Netherlands Organization for Scientific Research (NWO) (NWO-VIDI 864.13.013) to Jingyuan Fu and CardioVa sculair Onderzoek Nederland to Alexandra Zhernakova and Jingyuan Fu (CVON 2012-03) Alexandra Zhernakova holds an ERC-START grant (ERC-715772) and a Rosalind Franklin Fellowship from the University of Groningen. There was no additional external funding received for this study. The funders had no role in study design, data collection and analysis, decision to publish, or preparation of the manuscript.

==============================
Microbes in the gut microbiome form sub-communities based on shared niche specialisations and specific interactions between individual taxa. The inter-microbial relationships that define these communities can be inferred from the co-occurrence of taxa across multiple samples. Here, we present an approach to identify comparable communities within different gut microbiota co-occurrence networks, and demonstrate its use by comparing the gut microbiota community structures of three geographically diverse populations. We combine gut microbiota profiles from 2,764 British, 1,023 Dutch, and 639 Israeli individuals, derive co-occurrence networks between their operational taxonomic units, and detect comparable communities within them. Comparing populations we find that community structure is significantly more similar between datasets than expected by chance. Mapping communities across the datasets, we also show that communities can have similar associations to host phenotypes in different populations. This study shows that the community structure within the gut microbiota is stable across populations, and describes a novel approach that facilitates comparative community-centric microbiome analyses.

Introduction

The gut microbiome is a complex bacterial community, with its structure determined by many factors including the interactions between its members. Bacteria can interact in numerous ways—in either an actively targeted or passive manner, which can result in beneficial, neutral, or detrimental effects for the parties involved (Faust & Raes, 2012). Given the increasing evidence of the importance of the gut microbiome in human health, it is necessary to understand the inter-microbial effects underlying its composition.

Gut microbiota are frequently profiled using marker gene sequencing. Sequencing reads from amplicons of the selected marker are typically collapsed to operational taxonomic units (OTUs), analytical units used to approximate taxonomic abundances (Schloss et al., 2009; Navas-Molina et al., 2013). One approach to infer interactions between bacteria in the gut microbiota is to quantify the co-occurrence of OTUs across multiple samples (Faust & Raes, 2012). High correlation between OTUs can reflect the interactions between their source bacteria and similarities/differences in their responses to environmental conditions (Lozupone et al., 2012; Faust & Raes, 2012). However, OTU counts are relative to the sequencing depth of a sample, which introduces inherent correlations to the data (Friedman & Alm, 2012). As a result, several specialised approaches have been developed to estimate correlations from microbiota data (Faust et al., 2012; Friedman & Alm, 2012; Deng et al., 2012; Gevers et al., 2014; Fang et al., 2015; Kurtz et al., 2015). Whilst these have seen use within the research community (Gevers et al., 2014; Goodrich et al., 2014; Tong et al., 2013; McHardy et al., 2013), correlation metrics for microbiome studies have only recently been compared systematically by Weiss et al. (2016) who found that using an ensemble of metrics can improve the precision of co-occurrence detection.

Given co-occurrence between OTUs it is possible to generate networks of their inferred interactions. Within this we would not expect all bacteria to interact but rather that subsets of taxa are more likely to interact within one another forming identifiable and distinct interacting groups. We will hence refer to these sub-groups of co-occurring microbes as communities. The formation of such communities can be driven by factors such as cross-species metabolism and geospatial environmental variation (Faust & Raes, 2012; Levy & Borenstein, 2014; Lozupone et al., 2012). Previous studies have identified communities within microbial co-occurrence networks, often using the WGCNA (weighted gene co-expression network analysis) method developed to identify modules in gene co-expression networks (Lozupone et al., 2012; Tong et al., 2013; Jackson et al., 2016b; Duran-Pinedo et al., 2011; Langfelder & Horvath, 2008). However, whilst they may be more biologically accurate, such approaches allow OTUs to have weighted contributions to multiple communities, which complicates comparison and mapping of equivalent communities between networks. Using a community detection algorithm that assigns OTUs to single communities simplifies such analyses enabling comparison across datasets. This was demonstrated by the use of a modularity maximisation approach to detect and compare community structures between gut microbiome networks of irritable bowel syndrome patients and healthy controls (Baldassano & Bassett, 2016).

Figure 1 An overview of the study and the data used in analyses.

(A) An outline of the study design. (B) Plot of the first two components of PCoA from weighted UniFrac distance measures between samples in the study, coloured by dataset, shows there is some separation but significant overlap in microbiota compositions by study. (C) Comparison of alpha diversity measures in all three datasets. There was a significant difference in all pair-wise comparisons for both measures (Mann–Whitney U p < 0.0001). (D) Venn diagrams showing the number of OTUs shared across datasets. Showing both all OTUs and only those found in at least 25% of samples in each dataset. (E) Comparison of the mean relative abundances of taxonomies at the phylum and genus level across the complete table for each dataset. Phyla at less than 1% abundance and genera at less than 5% abundance in all sets are collapsed as Other. In genera f; and g; represent unassigned family and genus names in the Greengenes reference.

Here, we use the ensemble approach outlined in the recent methods comparison by Weiss et al. (2016) to quantify co-occurrence between gut microbiota, and apply a modularity maximisation approach to detect communities in the resultant networks. In this approach there are necessarily steps in which thresholds for edge inclusion and parameters for community detection must be selected. We describe biologically motivated and data driven approaches to inform these decisions. This method produces communities that can be compared and mapped across datasets. We apply this method to data from human cohorts from the UK, Netherlands, and Israel to establish whether gut microbiota form similar communities in different human populations. We find that OTUs form similar community structures across all three, and that these communities have similar associations with the health-related host factors of age and body mass index (BMI) in their respective populations. This study also provides a framework for future studies aiming to identify and, most importantly, replicate community level effects in microbiota studies.

Materials and Methods

Data aggregation

Given our objective to compare network community structures across datasets (Fig. 1A), we required OTUs that would be comparable between them. To this end, we carried out clustering of sequences across combined data from multiple sources. To maximise sequencing similarity, we selected two datasets with experimental approaches best matching those of the gut microbiota profiles obtained for TwinsUK, which used 16S rRNA gene sequencing of faecal samples. The LLDEEP and Israeli-PN datasets were selected as they carried out gut microbiota profiling by amplifying the V4 region of the 16S rRNA gene using the same PCR primers, and used paired-end sequencing on the Illumina MiSeq platform with read lengths sufficient to capture the whole V4 region. Notable differences between studies include faecal sampling and DNA extraction techniques. Both TwinsUK and LLDEEP utilised aliquots from faecal samples stored at −80 °C, whilst the Israeli-PN study utilised a mixture of faecal swabs stored at −80 °C and OMNIgene-GUT stool collection kits stored at −20 °C. All three studies used both chemical and mechanical lysis in DNA extraction but employed different protocols: TwinsUK utilised the MoBio PowerSoil HTP extraction kit, the LLDEEP cohort utilised the Qiagen AllPrep kit, and the Israeli-PN DNA was extracted using the MoBio PowerMag Soil DNA extraction kit.

Extraction of DNA from faecal samples, amplification and sequencing of the V4 region of the 16S rRNA gene, and demultiplexing of sequencing reads has previously been described for the TwinsUK cohort (Goodrich et al., 2014). The paired-end demultiplexed reads were joined using join_paired_ends within QIIME with an overlap of at least 200nt to form single reads covering the full V4 region (Navas-Molina et al., 2013). DNA extraction and 16S rRNA gene sequencing within the LLDEEP samples has been described in detail previously (Fu et al., 2015). Data from the LLDEEP cohort was provided in a similar format having used custom scripts to merge the paired end data to full length reads covering the V4 region and split data by individual (Gevers et al., 2014; Fu et al., 2015). Ethics approval for the TwinsUK study was given by the NRES Committee London—Westminster (REC Reference No. : EC04/015), and the Lifelines-DEEP study was approved by the institutional ethics review boards of the University Medical Center Groningen (ref. M12.113965).

Raw data accompanying the Israeli-PN publication was downloaded from the European Nucleotide Archive (ENA) (Accession: PRJEB11532) (Zeevi et al., 2015). This was processed similarly to the TwinsUK data and demultiplexed using published barcode mappings. Samples not listed in the accompanying metadata or with ambiguous read identifiers were removed.

OTU clustering, table filtering, and diversity analyses

The cleaned, joined, and demultiplexed data was concatenated to produce one sequencing file covering all three studies. After quality control, 4,426 samples (2,764 TwinsUK, 1,023 LLDEEP, 639 Israeli-PN) were included in the analysis. The complete sequencing from all three sets contained 381,767,528 reads. These were dereplicated, removing reads occurring only once, resulting in 5,728,288 unique reads. We chose to use a de novo approach to cluster these to OTUs, as de novo OTU clustering is not influenced by the reference database used, can capture novel diversity, and can produce more accurate clustering of OTUs than reference-based approaches (Westcott & Schloss, 2015; Jackson et al., 2016a). De novo clustering was carried out using the cluster_fast command in the VSEARCH package with a 97% similarity threshold (Rognes et al., 2016). The resultant 94,070 representative sequences were filtered to remove chimeric reads using the uchime_denovo command (within VSEARCH), producing a final set of 17,123 representative sequences. These were used to generate OTU counts across all samples using the VSEARCH usearch_global command (Rognes et al., 2016). Post-processing, average read counts were 82,695 ± 745 for TwinsUK, 49,962 ± 964 for LLDEEP, and 130,378 ± 5,534 for Israeli-PN (mean ± SEM).

A phylogenetic tree was generated from the representative sequences using the default parameters of the make_phylogeny command in QIIME (Navas-Molina et al., 2013). Taxonomy of OTUs was assigned by matching representative sequences against the Greengenes v13_8 database using the default parameters of the assign_taxonomy command in QIIME (Navas-Molina et al., 2013). OTUs occurring in only one sample, and samples with less than 10,000 OTU counts were removed. Weighted and unweighted UniFrac beta diversity measures and subsequent principle co-ordinates analysis of them was carried out using the beta_diversity_through_plots script in QIIME (Navas-Molina et al., 2013). The combined OTU table was then split by dataset. For the purposes of alpha diversity calculations, the raw counts tables were rarefied to a depth of 10,000 reads. For each sample, Shannon and inverse Simpson diversity indices were calculated as the mean across ten rarefactions. Significant differences in alpha diversity between datasets were assessed using Mann–Whitney U-tests.

Calculating co-occurrence measures

Co-occurrence calculation was carried out on each dataset independently. Sub-tables were generated from raw (unrarefied) OTU tables that only contained OTUs found in at least 25% of the samples (Jackson et al., 2016b). All co-occurrence measures were calculated within these subsets as they are less sparse and hence more amenable to correlation measures (Weiss et al., 2016). The mean diversity was assessed on rarefied versions of the subset tables using the inverse Simpson index as for the full tables. OTU sparsity was assessed from the unrarefied table using the biom summarize_table command. Following the recommendations of Weiss et al. (2016), we then used these estimates of the mean inverse Simpson index (TwinsUK = 20.2, LLDEEP = 29.0, and Israeli-PN = 13.1) and OTU table sparsity (0.49 in all three) to select an ensemble approach to co-occurrence detection that combines four different correlation measures: CoNet, SparCC, Pearson’s, and Spearman’s.

CoNet

CoNet is itself an ensemble approach. The package allows a range of co-occurrence measures to be combined with several options for how to combine the weighting of edges and their p-values. CoNet addresses compositionality within the data using the ReBoot procedure, which involves permutation followed by renormalisation of data. Calculating co-occurrence within these renormalized data allows assessment of the levels of correlation expected simply as a result of the compositionality within the data (Faust & Raes, 2012). For this study we used four measures of co-occurrence within CoNet: Spearman’s and Pearson’s correlations and Kullback–Leibler and Bray–Curtis distance measures (Faust & Raes, 2012). Initial correlation thresholds were selected for each of these measures that produced 2,000 positive and 2,000 negative edges concordant across the four metrics (Weiss et al., 2016); 1,000 permutations were then used for renormalisation to account for compositionality, and bootstrapping to identify edge p-values for each metric. The Simes method was selected to merge p-values across edges by keeping the minimum. Final p-values were adjusted for multiple testing using the Benjamini–Hochberg FDR approach.

SparCC

SparCC was developed to calculate correlations between OTU abundances in microbiome data whilst accounting for their inherent sparsity and compositionality (Friedman & Alm, 2012). It uses the centered log-ratio transformation to address data compositionality. SparCC was used with default parameters to calculate correlations from the raw count OTU tables (Friedman & Alm, 2012). The MakeBoostraps command was used to generate 100 bootstrapped tables, which were in turn used to calculate SparCC correlations. The bootstrapped correlations were then used with the PseudoPvals command to generate two-tailed p-values for the SparCC correlations from the true table.

Pearson’s and Spearman’s correlation coefficients

Pearson’s and Spearman’s correlation metrics do not take data compositionality into account, but were included to follow the approach outline by Weiss and Van Treuren et al. Both measures were calculated using the relative abundance tables. The rcorr function from the Hmisc R package was used to calculate correlations and generate two-tailed p-values pairwise between all OTUs for both Pearson’s and Spearman’s measures (Harrell Jr & Dupont, 2008). P-values were adjusted for multiple testing using the Bonferroni method in R, again following the approach of Weiss et al. (2016).

The outputs of all four co-occurrence approaches were converted into simplified unweighted edge tables detailing the direction of association (1 or −1) and bootstrapped/adjusted p-values.

P-value thresholding to generate co-occurrence networks

Intersected networks were generated by combining the edge tables from the CoNet, SparCC, Pearson’s, and Spearman’s methods. This was done independently for each dataset. Edges were retained in the final network if the direction of co-occurrence matched, and the edge p-values were below a given threshold in all four methods. This was carried out at multiple different p-value thresholds (0.05, then ranging in powers of ten from 0.01 to 10−8), to generate multiple intersect networks for each dataset with a gradient of stringency for edge inclusion. We then aimed to determine which was the most appropriate threshold to use to generate the final networks. Rather than make an entirely arbitrary selection, we chose to use fit to a scale-free network as a biologically motivated method to identify the optimum p-value threshold to use.

A scale-free network has a node degree distribution that follows a power law, i.e., there are few highly connected nodes and many more less connected nodes. This distribution has been observed in several biological networks, including microbiota co-occurrence networks (Jeong et al., 2000; Albert, 2005; Zhang & Horvath, 2005; Faust et al., 2012; Tong et al., 2013). It is also frequently used as a threshold for edge inclusion in the widely used WGCNA package (Langfelder & Horvath, 2008). Fit to a scale free network was calculated by first extracting the degree distribution of a network using igraph (Csardi & Nepusz, 2006). The fit of this distribution was then assessed using the scaleFreeFitIndex function from the WGCNA package in R (Langfelder & Horvath, 2008). This provides the R2 of the fit to a scale-free model, which the creators of WGCNA suggest should be >0.8, and the slope of the fit, which they suggest should be close to −2 to indicate a good fit. The optimum p-value threshold was selected based on these criteria across all three datasets (see Results) and the resultant intersect network at this threshold was used in all further analyses. Visualisation and generation of descriptive statistics from the final networks was carried out using Gephi (Bastian, Heymann & Jacomy, 2009).

Detecting communities within co-occurrence networks

Between OTU adjacency matrices were generated that represented the final intersect networks for each dataset. Negative correlations were removed (considered zeroes in the adjacency matrix) to generate unsigned networks as they represented a considerably small proportion of edges (<1% in all datasets). Community detection was carried out on each dataset’s network independently using the genLouvain 2.0 package within MATLAB (Mucha et al., 2010), which implements the Louvain approach to modularity maximisation (Blondel et al., 2008). This defines community partitions by assigning nodes to unique communities, then iteratively combining neighbouring nodes into communities if it results in an increase in modularity across the whole network. Modularity is a measure of the number of edges within communities relative to between communities and a higher value represents better community definition (Newman & Girvan, 2004).

The genLouvain algorithm includes a γ parameter, which controls the size and number of detected communities (Mucha et al., 2010). A smaller gamma value promotes the detection of a small number of larger communities, while larger gamma values promote the detection of a high number of smaller communities. To find an optimal value for the γ parameter, we carried out community detection using a range of γ values for each dataset (0.1–1, increments of 0.01), and assessed the stability and statistical significance of the resultant communities.

Stability

To assess the stability of community definitions at each γ, we carried out community detection 25 times on the real network followed by pairwise comparisons of the similarity of community clustering between the runs. To assess similarity between two community groupings we used the normalised variation of information (variation of information divided by ln(number of nodes in the network)) as a measure of similarity between assignments (Ronhovde & Nussinov, 2009). A high value for variation of information means OTUs are grouped more differently in the two partitions compared, whereas a value of zero means the two partitions are identical. In figures and text, we report 1-normalised variation of information so a higher value represents more similar community structure.

Statistical significance

To assess the statistical significance of the community definitions, we also carried out community detection at each γ on 100 randomised networks with nodes following the same degree distribution as the real network (generated using the randomGraphFromDegreeSequence command in the Octave networks toolbox package). We then compared the mean modularity of the 25 runs on the real network to the 100 randomised networks. A process that has previously been proposed to assess the significance of community structures across different resolutions (Lambiotte, 2010; Traag, Krings & Van Dooren, 2013).

From observation of both the variation of information (stability) and modularity (statistical significance) results we then selected a suitable value for γ that produced both stable and significant community groupings (see Results). Once a suitable value for γ was identified, community detection was repeated 100 times at this value, and the community definitions from the run producing the highest modularity were retained as the final community definitions.

Community properties and host associations in TwinsUK

The relative abundance of each community in a sample was found by summing the read counts of its constituent OTUs and dividing by the total number of reads observed in the sample. Association between the mean abundance of an OTU in a dataset and the number of OTUs in its parent community was assessed using Spearman’s correlation. Taxonomy within communities was investigated by counting the number of OTUs assigned to different taxa at each taxonomic level. The mean identity between the representative sequences of the OTUs in each community was assessed for each dataset. All pairwise comparisons between the OTUs within a community were carried out using global alignments performed using BLAST within the pairwise2 command of Biopython, with a score of 1 for a match, −1 for a mismatch, −2 for opening a gap, and −1 for extending a gap (the defaults used by the Mothur align.seqs command) (Cock et al., 2009; Schloss et al., 2009).

The heritability of TwinsUK communities was estimated using the mets package in R. Log transformed relative abundances were used for each community. For each community ACE, CE, E, and AE models were fit using data from complete twin pairs. Co-variates included age, BMI, sample collection method, sex, and sequencing depth. For each community, the best fitting model was determined as that with the lowest Akaike information criterion.

Association analyses between community abundances and BMI and age was carried out using log transformed relative abundances of the communities (log10 (relative abundance +10−6)). These phenotypes were selected as they were also available in both the LLDEEP and Israeli-PN datasets. To investigate associations, linear models were fitted for each community using the lm function in R. Community abundance was the dependent variable with BMI, age, gender, and sequencing depth as independent variables, as these were the maximal set available across all datasets. The coefficient and significance of associations were extracted for BMI and age. P-values were then adjusted for multiple testing using the FDR method in R.

Comparison of community structure between populations

Overall community structure comparisons

We carried out pairwise comparisons between datasets to assess the similarity of overall community structures between the networks. In each comparison, the two networks were subset to just the OTUs shared by both, and the normalised variation of information between their community structures calculated. To find the variation that would be expected by chance, we generated randomised community sets for each network by shuffling OTU labels. Each randomised comparison therefore shared the same number OTUs between the two real networks with the same community sizes, but without the biological basis for the linkage of OTUs. We then carried out pairwise comparisons of the variation of information between the randomised communities. Shuffling and comparisons were repeated 1,000 times. The highest score observed (1 −normalised variation of information) in any pairwise comparison, in any permutation was 0.41 (mean = 0.34, SD = 0.014).

To determine the robustness of the community detection approach to differences in the selected parameters, overall community structure was also compared between networks created using both higher and lower p-value thresholds and γ values. In each case, networks and community definitions were generated as previously. Normalised variation of information was then used to compare the networks’ community structures as described above. Comparisons between the three datasets were carried out within each p-value and γ pairing. A wider comparison was also carried out between all possible dataset, p-value, and γ value combinations. In the case of the complete pairwise comparison, permutations to assess null variation of information were only generated 100 times (in place of 1,000) due to the high total number of comparisons involved.

Mapping of individual communities between datasets

To map communities between networks we carried out pairwise comparisons between all the communities in all three networks. We used the Jaccard index to quantify the number of OTUs shared between the two communities relative to the number not shared in each comparison. Matches were considered positive with scores >0.25 (range 0–1, no shared OTUs—complete overlap). This was selected as above this threshold there were no instances of multiple mapping of communities between datasets.

Community-types were defined where matches could be found linking the communities in all three datasets. These were labelled with colour names using the standardColors function from the WGCNA package in R. The log transformed abundances for the 14 community-types that could be mapped across all three datasets were generated for the LLDEEP and Israeli-PN datasets as for TwinsUK. These were analysed for associations with age and BMI using linear regressions as for TwinsUK.

Results

Integrating microbiota data from different populations

The aim of this study was to compare gut microbiota community structures across populations (study overview in Fig. 1A). To this end comparable OTUs were generated by combining gut microbiota sequencing data from three geographically diverse populations. These were the British TwinsUK and Dutch Lifelines-DEEP (LLDEEP) cohorts and data from an Israeli personalised nutrition study (Israeli-PN) (Table 1) (Fu et al., 2015; Zeevi et al., 2015). Principle coordinates analysis of overall microbiota composition between samples found that whilst there was some grouping by dataset, there was also significant overlap between them (Fig. 1B). Comparison of median alpha diversities between the populations found significant differences between all three (Fig. 1C).

Table 1 Participant summary statistics for the datasets considered in this study.

Dataset	BMI (Mean ± SD)	Age (Mean ± SD)	Sex (M/F/unknown)	No. samples	
TwinsUK	26.1 ± 4.8	59.5 ± 12.3	308/2,456/0	2,764	
LLDEEP	25.3 ± 4.2	45.3 ± 13.7	445/578/0	1,023	
Israeli-PN	26.4 ± 5	43.5 ± 12.9	376/251/12	639	

Across the 15,361 OTUs detected in the complete data 48% were shared across all three datasets, with 79% being found in at least two datasets (Fig. 1D). There was also considerable overlap between TwinsUK and LLDEEP of OTUs not found in the Israeli-PN data. Similar patterns were observed when considering more abundant OTUs (found in >25% of the respective populations). Although LLDEEP and TwinsUK shared more OTUs, examining the mean taxonomic distributions at the phyla level the TwinsUK and Israeli-PN cohorts were most alike (Fig. 1E). The LLDEEP cohort contained relatively lower levels of the phylum Bacteroidetes, and a higher abundance of Firmicutes bacteria. Further differences were observed at the genus level, where the Israeli-PN study had a higher average abundance of Prevotella.

Scale-free p-value thresholding of edges in co-occurrence networks

We split the OTU data and generated co-occurrence networks for each dataset using an ensemble approach—taking the intersect of four different correlation measures (TwinsUK example in Fig. 2A). This required selection of a p-value threshold at which correlations were considered significant and retained as edges in the networks. We generated ensemble networks for each dataset at a range of p-value thresholds and selected a cut-off where the resulting networks’ degree distributions best fit a scale-free distribution (Fig. 2B). We observed no consistent trend across the p-values, but this might be expected given the differences in the number of OTUs and samples in the datasets and varying levels of p-value precision between the co-occurrence measures used. Indeed, investigating network properties at each threshold for each method individually (Fig. S1), we observed that the p-value accuracy of SparCC meant that at at all thresholds below 10−2 there was no change in the number of edges in the SparCC networks (only edges with p = 0 remained). We also observed that the drop in fit to a scale-free distribution at lower p-value cut-offs in the TwinsUK and Israeli-PN networks was driven by trends in the CoNet networks, which had the lowest number of edges and hence were the principal determinant in the intersects at these lower thresholds. However, overall, the trends of the ensemble networks did not simply reflect those of any one of their constituent methods, displaying properties only emergent upon intersection of all four. From examination of the ensemble networks (Fig. 2B), we found that including edges with a p-value <0.01 produced intersect networks with a good fit to the scale-free model in all three datasets. As such, this threshold was selected to generate the networks used for community detection.

Figure 2 P-value thresholding of co-occurrence networks.

(A) A visualisation of the ensemble process at the p < 0.01 threshold for TwinsUK. The Pearson, Spearman, CoNet, and SparCC networks combine to make the final Intersect network. (B) Selection of final p-value threshold for generating ensemble networks. Networks for each dataset were made by intersecting four correlation measures with different p-value thresholds. The R2 and slope of the resultant intersect networks regression fit to a scale-free distribution was assessed using WGCNA, highlighted in red are the developers recommend values to consider a good fit. Also shown are summary measures of the resultant networks at each value showing the mean node degree of OTUs in the networks and the total number of edges in each. The black dashed line indicates the final p-value threshold chosen to generate networks for community detection.

From the 6,761 unique edges observed across in all three networks, 166 were observed in all three, 882 in the TwinsUK and LLDEEP, 677 in TwinsUK and Israeli-PN, and 229 in the LLDEEP and Israeli-PN datasets. Summary statistics for the resultant networks are shown in Table 2. The TwinsUK and LLDEEP network structures were most alike. The Israeli-PN data, which contained fewer nodes (OTUs), produced a smaller and more connected network.

Table 2 Summary statistics for the final intersect co-occurrence networks (p < 0.01) for each dataset.

Graph density is the percentage of all possible edges represented, mean path length is the minimum calculated pairwise between all nodes, mean clustering coefficient is across all nodes in the network and provides a comparative indicator of overall clustering in the networks.

Dataset	Nodes	Edges	Mean node degree	Graph density	Mean path length	Mean clustering coefficient	
TwinsUK	844	2,843	3.3	0.008	5.9	0.29	
LLDEEP	922	2,967	3.2	0.007	5.9	0.28	
Israeli-PN	406	2,573	6.7	0.033	3.7	0.31	

Detection of communities in microbial co-occurrence networks

Selecting a γ parameter for community detection

After establishing co-occurrence networks we used the Louvain modularity maximisation algorithm to detect communities within them (Mucha et al., 2010; Blondel et al., 2008; Newman & Girvan, 2004). The version used includes a constant (γ) that can be used to manipulate the number and size of resultant communities (see ‘Materials and Methods’). To determine an appropriate value for γ, we carried out repeated community detection on the three co-occurrence networks at various γ values and assessed the stability (as variation of information between runs) (Fig. 3A) and statistical significance (mean modularity of real compared to randomised networks) (Fig. 3B) of the community definitions at each γ.

Figure 3 Selecting a γ parameter for Louvain thresholding.

Louvain community detection was carried out multiple times on each network on a range of γ values. (A) The mean variation of information between the 25 runs at each gamma value. Plotted is 1-variation of information so a higher value means more stability in community definitions between runs. (B) The mean modularity of the 25 iterations compared to the mean modularity of 100 randomised networks with the same degree distribution. (C) The mean number of communities generated at each value of γ. In all plots, the mean line is surrounded by the interval of standard deviation. This is also true for the plots in B, but it should be noted that modularity estimates had extremely low variance between runs, hence even a small difference between the random and true means is significant. The dashed red line indicates the final value selected for γ.

Community definitions were stable (low variation of information between runs) at γ values approaching zero (Fig. 3A). This would be expected as in this instance most OTUs fall into a few large communities (Fig. 3C). Similarly, at these low γ values, a networks edges are most likely to be within communities, hence we observed high modularity values for both the true and randomised networks. As γ increased we found modularity estimates for the random networks dropped significantly lower than those of the true networks (Fig. 3B). This shows that the real co-occurrence networks have significantly more community structure than would be expected by chance, and provides confidence that the communities identified at these γ values reflect the biological relationships of their member OTUs. The stability of definitions (mean variation of information) fluctuated across the γ range, however it should be noted that the lowest estimates (1—variation of information) were above 0.9 (range 0–1, least to most similar), showing that even the least stable γ produced very similar communities between runs.

Final community definitions

From Fig. 3, we selected a γ value of 0.4 for community detection. This γ provided both high modularity estimates (that were significantly higher than the random networks) and good stability in all three datasets. We then used the Louvain algorithm with γ = 0.4 to detect the final OTU community definitions in the three co-occurrence networks (Fig. 4A, Table S1). Communities in each network are hence referred to using arbitrary numbers ranging from 1 to the number of communities in the network.

Figure 4 Communities detected within the co-occurrence networks.

(A) A visualisation of the communities detected by the Louvain algorithm. Communities are arbitrarily coloured. Node size represents an OTUs relative abundance within the dataset. (B) Summary of the taxonomic distributions within the TwinsUK communities. Only communities with at least 5 OTUs are shown. Stacked bars represent the number of OTUs in the community assigned to each taxon. (C) Community plot for TwinsUK as in A, but coloured by the heritability estimate of the community. (D) Highlighting the significant negative association between BMI and the abundance of a Christensenellaceae rich community (linear regression with age, gender, and seq. depth β =  − 0.13, FDR q = 0.001).

The number of communities detected within the networks (TwinsUK = 96, LLDEEP = 105, Israeli-PN = 31) reflected the number of OTUs within them, but was more similar when only considering communities containing at least five OTUs (TwinsUK = 35, LLDEEP = 36, Israeli-PN = 11). There was a positive correlation between the mean relative abundance of an OTU and the total number of OTUs in the community it was assigned to in the TwinsUK and LLDEEP networks (rs = 0.1, p = 0.002 and rs = 0.27, p < 0.0001 respectively). However, overall each community in all three networks contained a range of both high and low abundance OTUs (Fig. S2).

Taxonomy, heritability, and host associations of communities in TwinsUK

Taxonomic distribution within communities

Investigating taxonomic assignments of OTUs within the TwinsUK communities, we found that at finer taxonomic levels (genus and species) several communities contained a mixture of OTUs assigned to different taxa (Table S2). Whilst some communities retained a mixture of taxa at the family level, the majority were dominated by one taxon (Fig. 4B), and at higher levels nearly all contained a single taxon. A similar pattern was observed in the LLDEEP and Israeli-PN networks (Fig. S3). These results also reflected the mean sequence similarity between OTUs within communities.

We quantified the average pairwise distance between the representative sequences of the OTUs within each community within each dataset (Fig. S4 and Table S3). We found that most communities had a mean sequence identity of 0.94–0.97 between their OTUs. This further suggests communities largely consist of OTUs from taxonomically similar sources. However, we also observed communities with higher divergence between the representative sequences of their OTUs. These included communities 1 (0.84) and 3 (0.82) from the Israeli-PN network, communities 4 (0.82) and 19 (0.84) from the LLDEEP network, and communities 5 (0.88) and 26 (0.88) from the TwinsUK network amongst others (mean identity between OTU representative sequences).

The influence of host genetics on communities

To determine if host genetics influenced the abundance of the microbial communities in TwinsUK, we used an ACE model (which estimates the variance attributable to additive genetics—A, to the environment common within twin pairs—C, and to the environment unique to individuals—E) to estimate the heritability of community abundances within 654 monozygotic and 495 dizygotic pairs. From the 96 communities in the TwinsUK network 52 had some variance attributed to additive genetic effects (Table S2). Within these, the variance due to genetic effects ranged from 0.07–0.46 (Fig. 4C). The estimates for the most heritable communities are of a similar magnitude to those of the most heritable taxa previously reported within TwinsUK (Goodrich et al., 2014). Indeed, examination of taxonomies within these communities found that they contained highly heritable taxa (Table S2). For instance, the most heritable community (Community 39) contained a Turicibacter OTU, and the second most (Community 26) contained numerous Christensenellaceae and Ruminococcaceae OTUs (Goodrich et al., 2014).

Community associations with age and BMI

To further explore host effects, we investigated if communities were associated with properties relating to health. We carried out linear regression analysis between community abundances and age and BMI in TwinsUK (Table S4). These phenotypes were selected as they were available for all three datasets enabling replication. In TwinsUK, we found that of the 96 communities 47 were significantly associated with BMI (FDR adjusted p < 0.05). The highly heritable Community 26 that contained multiple Christensenellaceae OTUs had a significant negative correlation with BMI (FDR adjusted p < 10−10, β =  − 0.13) (Fig. 4D), reflecting our previous observation that Christensenellaceae in conjunction with correlated taxa are protective against weight gain in mice (Goodrich et al., 2014). There were several communities containing exclusively OTUs belonging to the order Clostridiales that were negatively associated with BMI. The strongest positive association was with Community 5 (FDR adjusted p < 10−7, β = 0.11), which was a large community also dominated by Clostridiales OTUs. However, these were also assigned lower level taxonomies with the majority being various genera from the Lachnospiraceae family.

There were 48 communities significantly associated with age (FDR adjusted p < 0.05). The three most significant associations were negative, two of these (Community 54 and 24), were dominated by Bifidobacterium OTUs. There was also a significant negative association with Community 27 (FDR adjusted p = 0.007, β =  − 0.05), containing Faecalibacterium prausnitzii OTUs. The strongest positive association with age was observed with Community 17 (FDR adjusted p < 10−7, β = 0.11), which contained exclusively Enterobacteriaceae OTUs. There were also several small communities positively associated with age that contained exclusively OTUs assigned to the Ruminococcaceae family.

Comparison of communities across populations

Similarities in overall community structure

Having identified communities within each dataset we aimed to determine how similar the segmentation of OTUs was between them. We again used the normalised variation of information measure to compare groupings, and found that the similarity of community assignments was significantly higher in all three pair-wise comparisons than would be expected by chance (Fig. 5A). This shows that OTUs are forming similar communities across the three geographically diverse datasets.

Figure 5 Comparison of communities across populations.

(A) Pair-wise comparison of variation of information between community definitions for OTUs shared between networks. 1-variation of information is shown where 1 represents identical segmentation of OTUs to communities and 0 indicates no similarity, with the highest score observed in randomised permutations being 0.41. Numbers represent the number of OTUs shared between the sets (OTUs with at least one edge). (B) A network showing the matching of communities based on Jaccard index. Edges are considered where the index >0.25 and are weighted by the index 0.25–1. Highlighted are the community-types selected for later analysis, using the colour matching their name. (C) Visualisation of networks as in 4A but coloured based on community-types as in B. Communities not in these 14 types are coloured grey. (D) Replication of linear regression analysis for each of the 14 mapped communities in the LLDEEP and Israeli-PN for associations with both age and BMI. Squares are coloured by the Beta estimate for the association and nominally significant associations (p < 0.05) are highlighted by a black border. Community-types are ordered by their associations within TwinsUK.

We also repeated community structure comparisons between networks using both higher and lower p-values to threshold edges, and higher and lower values for γ in community detection (Fig. S5 and Table S5). In all instances the community structure was higher than expected by chance, and similar communities were observed both within and between datasets when using alternate parameters. This shows that the approach is robust to differences in parameterisation, and that the similarity observed between datasets is not solely manifest at the selected values. However, it should be noted that in all comparisons the variation of information can only be used to compare matching sets, so these results only show communities are similar within the OTUs shared by networks.

Mapping equivalent communities between datasets

We mapped the communities (those defined using γ = 0.4 and a network edge threshold of p < 0.01) between the three networks using the Jaccard index to identify those which were equivalent to one another. Across the 96 TwinsUK, 105 LLDEEP, and 31 Israeli-PN communities, we found 14 instances where communities could be matched in all three datasets (Fig. 5B), although there were many more that could be mapped across only two datasets. For distinction, we refer to these 14 matched groups as community-types and label each using arbitrary colour names (Figs. 5B and 5C, Table S6). For example, the Green community type was community 45 in the LLDEEP network, community 31 in the Israeli-PN network, and community 36 in the TwinsUK network.

Communities have similar associations with age and BMI across populations

To determine if each of the 14 community-types had similar associations with age and BMI in their respective populations, linear regressions were carried out as for TwinsUK. Comparing the results (Fig. 5D), we found that seven significant BMI associations within TwinsUK were replicated in the LLDEEP data, with two also significant in the Israeli-PN data. There were six community-types significantly associated with age in both the TwinsUK and LLDEEP data, four of which were also significant in the Israeli-PN data. BMI had more negative associations within the community-types, whereas age had more positive associations. For both BMI and age, the significance of community associations was most similar between TwinsUK and LLDEEP, this is may be due to their higher sample sizes relative to the Israeli-PN dataset.

The Greenyellow and Green community-types had significant negative associations with BMI in all three datasets. Greenyellow communities were dominated by Ruminococcus OTUs in all three, and similarly all the Green communities consisted solely of Ruminococcaceae OTUs. More widely, most community-types with at least one significant negative association with BMI consisted of Clostridiales OTUs. The Pink community type for instance had a significant negative association with BMI in TwinsUK and LLDEEP, and contained multiple Lachnospiraceae OTUs assigned to the species Coprococcus eutactus in all three datasets.

The Salmon, Turquoise, Cyan, and Red community-types were significantly positively associated with age across all three datasets. All three contained exclusively Clostridiales OTUs. The strongest positive association with age in TwinsUK was with the Brown module which was not associated with age in the other sets, this contained Veillonella and Haemophilus parainfluenzae OTUs. Several associations for both age and BMI were significant in TwinsUK and LLDEEP but not in the smaller Israeli-PN dataset, but tended to share the same direction of effect. Overall, these results showed that communities associated with age and BMI in the same manner across the thee populations.

Discussion

Here, we have presented a rationalised and data-driven approach to generate co-occurrence networks from 16S rRNA gene sequencing data, and to identify comparable community structures within them. Applying this to gut microbiota profiles from three different populations we have found that both co-occurrence networks and community structures are stable across all three. Furthermore, we have shown that genetics, age, and BMI are associated with the relative abundances of the identified communities, with age and BMI having similar associations with communities in all three populations.

Differences between datasets

In combining these datasets we observed significant differences in the diversity and taxonomic profiles between datasets. These likely reflect a mixture of both the differences in the study populations and the experimental approaches used. Environmental differences between cohorts, in terms of genetics and other lifestyle factors are known to influence the gut microbiome (Goodrich et al., 2014; Falony et al., 2016; David et al., 2014). For instance, low diversity Prevotella dominant microbiomes have been previously observed in cohort studies at differential levels and are potentially linked to dietary intake (Arumugam et al., 2011; Wu et al., 2011). The larger proportion of Prevotella dominance, and associated reduced diversity, in the Israeli-PN gut microbiota might therefore reflect geographical and dietary differences. However, technical disparities such as faecal sampling and extraction method, which were known to be different between the present studies, are also known to influence taxonomic composition and cannot be discounted (Walker et al., 2015; Kennedy et al., 2014; Sinha et al., 2017). The increased overlap in the TwinsUK and LLDEEP data in comparison to the Israeli-PN dataset in both the OTU and network structures might also result from the higher sample sizes in these studies. However, it is notable that even with the intrinsic population and technical differences between the three studies we still observed consistent elements across them. Most OTUs were found in at least two studies and there was overlap of samples from all three in beta diversity PCoA plots. It is within this common ground that similarities in community structure were observed.

Generating co-occurrence networks

To generate the co-occurrence networks in this study we used an ensemble approach as suggested by Weiss et al. (2016). However, we found that CoNet was the main edge filter when intersecting networks and also produced the best fit to a scale-free distribution when considered alone, and thus might be sufficient. This might be expected given that it is itself an ensemble approach. However, metrics such as SparCC are not implemented within CoNet, and different correlation methods can produce better results depending on the format and properties of the input dataset (Weiss et al., 2016). These observations may therefore only apply to the current study. Furthermore, there are several alternative approaches that were not considered in the comparisons of Weiss and Van Treuren et al. such as SPIEC-EASI and CCLasso (Fang et al., 2015; Kurtz et al., 2015). Further exploration of community detection using these methods is warranted. An approach to generate quantitative microbiota profiles using a sample’s total microbial cell count has also recently been described (Vandeputte et al., 2017). This negates issues of data compostionality and was shown to reduce the number of spurious correlations observed between microbiota. Using such approaches in future studies could enable the use of more traditional correlation metrics and simplify downstream community detection.

We selected edges for inclusion in co-occurrence networks based on their fit to a scale-free topology. Comparing community structures using different p-value thresholds we observed similar community structures between different datasets when using different thresholds. This suggests different p-value cut-offs could be used for each dataset if a single threshold does not produce a reasonable scale-free fit across all datasets as in this study. However, we also observed similar community structures within datasets when using thresholds producing networks with a poorer fit to a scale-free distribution. This is likely due to the use of a more stringent ensemble approach to co-occurrence estimation, which limits edges in the network to the strongest and most consistent associations (arguably most likely to reflect biological phenomena) prior to further thresholding. Use of a scale-free approach to p-value parameterisation might therefore be unnecessary where similar stringent co-occurrence methods are used. It is also not possible to determine if the true network has a scale-free distribution, and indeed the true network could be defined in a number of different ways depending on what relationships are considered to constitute an interaction or edge.

However, it has previously been shown that microbial interaction network structures can approximate a scale-free distribution (Chaffron et al., 2010; Faust et al., 2012; Tong et al., 2013). Scale-free networks are also well conserved in other aspects of biology across different domains of life (Wolf, Karev & Koonin, 2002; Albert, 2005), and this approach is used by existing methods such as WGCNA (Langfelder & Horvath, 2008). Using typical co-occurrence approaches a p-value cut-off will be required to determine edges for inclusion when generating a network. Using a scale-free fit, or similar data driven approach, avoids using entirely arbitrary selections and provides a uniform rationale that can be used to generate comparable network structures across different datasets.

Community detection

Following network creation, we utilised a modularity maximisation algorithm to detect communities within the network. Comparing modularity to randomised networks ensured we were detecting structure manifest as a result of non-stochastic processes. Similarly, our application of variation of information enabled identification of the most stable communities across the range of γ values tested. Such parametrization steps are often neglected in network analyses but should be considered in future microbiota studies, particularly where the aim is to identify the optimal community configuration. However, overall the community definitions were stable and similar across the range of γ values considered here, suggesting that the resultant OTU communities were robust to small changes in the γ parameter.

Modularity maximisation assigned OTUs to single communities. This may not represent the true nature of microbiota interactions but provides some analytical benefits. Unambiguous definition of communities facilitates comparison between network structures. This has applications in comparison between disease and control groups (Baldassano & Bassett, 2016) or, as we have shown, replication of community associations across datasets. Unambiguous assignments of OTUs to individual communities might also be desirable for studies aiming to design synthetic communities to replicate the benefits observed with treatments such as faecal microbiota transplants (De Roy et al., 2014). For example, these methods could be applied to identify common microbial communities in successful donor samples, and generate more controlled, rationally designed synthetic communities for use in place of current less specific approaches. There is also evidence that a community-centric approach to studying the gut microbiome in relation to human health can be more relevant than investigating individual taxa. Several studies have shown that associations between individual gut microbiota and host health can depend on their wider community context (Gevers et al., 2014; Goodrich et al., 2014; Baldassano & Bassett, 2016; Ridaura et al., 2013).

Beyond identifying communities, understanding their formation would require extension of the described approaches to quantification of taxa using metagenomic sequencing. This, especially used in tandem with metatranscriptomics and metabolomics, might provide indications to the mechanisms of interaction. Metabolic modeling from metagenomic data can also be used to improve the inference of interactions from co-occurrence observations by predicting interspecies metabolic dependencies and/or shared niche specialization (Levy & Borenstein, 2013). However, even with such improvements, cross-sectional approaches are inherently limited to inference of interactions from co-occurrence across samples. Time-series data and in vitro studies will also be required to delineate directional effects and validate individual interactions (Faust et al., 2015). For instance, there are existing methods that have been used to infer interactions from covariation between microbiota across time series (Steele et al., 2011), and it has been shown that interactions observed in pair-wise species co-cultures can be used to predict outcomes in more complex multi-species cultures (Friedman, Higgins & Gore, 2017). Combining in vitro observations with community interactions observed in the host environment could be a powerful tool for the design of synthetic bacterial communities for use as gut microbiome targeting medicines (Lindemann et al., 2016).

The stability of communities across populations

Although 16S rRNA gene data cannot elucidate interaction mechanisms, we were able to determine several biological phenomena from the microbial approximations provided by OTUs. Most notably, that OTUs formed similar communities with similar host associations in the co-occurrence networks of three different populations. This shows that the variation that exists between the populations (e.g., the OTUs unique to each dataset) does not significantly alter the interactions between the OTUs shared across all three.

Several community associations with age and BMI in TwinsUK replicated in the LLDEEP and Israeli-PN datasets. These associations with age and BMI, and the heritability results within TwinsUK, broadly reflected previous observations from studies investigating taxa in isolation. For instance, in TwinsUK communities negatively associated with BMI were enriched with butyrate producers Ruminococcaceae and Coprococcus eutactus (Louis & Flint, 2009). Members of the Ruminococcaceae family have previously been associated with visceral fat mass in members of the TwinsUK cohort (Beaumont et al., 2016). Negative associations with butyrate have also been previously observed with metabolic deficits such as type 2 diabetes that are also associated with obesity (Gao et al., 2009; Qin et al., 2012). However, short chain fatty acids have also been observed at higher levels in obese mice (Turnbaugh et al., 2006), and specific taxa associations with obesity were not found in a recent meta-analysis of human data using BMI (Sze & Schloss, 2016). We observed two Bifidobacterium rich communities that were negatively associated with age. This is in line with two previous studies (Yatsunenko et al., 2012; Odamaki et al., 2016), although it should be noted that these considered a wider range of ages and the principal loss of Bifidobacterium with age was observed in infants. We also observed F.prausnitzii communities that were negatively associated and a community of Enterobacteriaceae that was positively associated with age, similar to previous observations within TwinsUK with frailty (Jackson et al., 2016b). Furthermore, the most heritable communities contained OTUs belonging to highly heritable taxa (Goodrich et al., 2014).

Co-occurrence patterns between taxa

It would be expected that community level associations reflected those of taxa based studies as each community constituted of taxonomically similar OTUs. This is in agreement with a previous study that found that co-occurrence to be higher between genetically similar taxa (Chaffron et al., 2010). This indicates either that interactions might evolve within closely related taxa; or that co-occurrence mainly detects genetically related taxa (more likely to have similar functionality) responding to environmental stimuli in the same manner. Reflecting observations that niche differentiation can be a major driver of co-occurrence patterns (Levy & Borenstein, 2013). A further possibility is that co-occurrence communities are grouping reads from source taxa that are improperly captured by the heuristic sequence clustering used to generate OTUs. Whilst this is unlikely the main driver of these communities (most had mean sequence identities more divergent than the 97% threshold), further exploration is warranted to determine the influence of OTU clustering threshold on co-occurrence patterns and communities.

We also observed several more taxonomically diverse communities with higher levels of sequence divergence between OTUs. From those highlighted in the results: The Israeli-PN community 3 contained a diverse number of taxa that have been associated with short chain fatty acid (SCFA) production including the genera Coprococcus and Blautia and the Rikenellaceae family (Duncan et al., 2002; Vital et al., 2015; Louis & Flint, 2017). Similarly, numerous taxa in Community 1 from the Israeli-PN network have been associated with SCFA production, including F.prausnitzii, Roseburia, Blautia, and potentially Oscillospira (Sokol et al., 2008; Louis & Flint, 2017; Konikoff & Gophna, 2016). Community 4 from the LLDEEP network also contained a similar combination of OTUs assigned to these same SCFA associated taxa. The LLDEEP community 19 contained 29 OTUs all assigned to the order Clostridiales but with a mixture of finer taxonomic assignments including Oscillospira and the family Christensenellaceae, which, as well as being a highly heritable taxon, has been experimentally shown to influence Oscillospira abundance in mice (Goodrich et al., 2014). OTUs within Community 5 from the TwinsUK network also contained taxa related to SCFA production including Blautia and Coprococcus, and TwinsUK community 26 contained Christensenellaceae and Oscillospira OTUs, similar to LLDEEP Community 19.

These results suggest that the communities containing a more diverse range of taxa are largely driven by a shared involvement in short chain fatty acid production in the gut. This could be due to each taxa having similar responses to host environment and substrate supply, or due to metabolic inter-dependencies between them. Further work using metabolic and metagenomic data will be required to determine which is the case, and is warranted given the beneficial health effects associated with gut microbial SCFA production (Maslowski & Mackay, 2011). The conserved communities containing Christensenellaceae and Oscillospira are also of interest given their established interaction and the heritability of the former family (Goodrich et al., 2014). Stable observation of this community across human cohorts might reflect a direct host action on the gut that selectively promotes these communities.

We are unable to determine if the co-occurrence communities observed here reflect closely related taxa responding to environmental niches in a similar manner, or direct interactions between microbes (such as metabolic dependencies) driving inter-dependencies between them. Most likely it is a combination of these factors. Nevertheless, co-occurrence communities provide a useful method to reduce the dimensionality of 16S rRNA gene sequencing data to units reflecting biological phenomena. Further analyses using the described approaches should investigate the influence of host factors such as genetics, diet and health, on these communities and their reciprocal influences on the host.

Conclusion

We have described a method to generate robust and comparable community definitions from microbiota co-occurrence networks. We have also described data-driven parameterisation steps and methods to map communities and compare their associations across datasets. This enabled us to demonstrate that the gut microbiome contains stable communities of bacteria that are similarly associated with host factors across geographically diverse populations. Future use of this approach will facilitate community-centric microbiota studies, in particular by aiding replication of findings across datasets.

Supplemental Information

Supplemental Information 1 Supplementary Methods

An overview of code and scripts used in main stages of network creation and community detection.

Click here for additional data file.

Supplemental Information 2 Network fits to a scale-free node distribution and other network statistics across different p-value thresholds, for each dataset, split per co-occurrence calculation method

Model fits and statistics were derived using igraph and WGCNA as for the full intersected networks. Mean path length refers to the mean shortest path length between all pairs of nodes.

Click here for additional data file.

Supplemental Information 3 A comparison of the relative abundance of an OTU to the number of OTUs in its assigned community for each of the three data sets

Click here for additional data file.

Supplemental Information 4 Taxonomic summary plots for the communities in the LLDEEP and Israeli-PN data, as for TwinsUK (Fig. 4B)

Click here for additional data file.

Supplemental Information 5 Mean sequence identity between the representative sequences of the OTUs in each community

Shown for all communities with greater than two OTUs. The dashed line represents the 0.97 threshold used to cluster OTUs.

Click here for additional data file.

Supplemental Information 6 Community structure comparisons pairwise between all community defintions generated across all datasets, p-values, and gamma values

Variation of information was used to compare each community definition pairwise. For each pairwise comparison the two definitions were shuffled 100 times to generate null estimates. The maximum null estimate observed across all permutations from all comparisons is highlighted on the scale as is the minimum real value observed. Numbers on the points represent the number of overlapping OTUs between the networks used in the comparison.

Click here for additional data file.

Table S1 The OTUs assigned to each community across all three datasets

Click here for additional data file.

Table S2 Heritability analysis results for the communities in TwinsUK

Click here for additional data file.

Table S3 Mean sequence identity between the representative sequences of OTUs in each community

Click here for additional data file.

Table S4 Community association results with age and BMI in TwinsUK (Community members can be found in Table S1)

Click here for additional data file.

Table S5 Maximum null and minimum real normalised variation of information values observed when comparing the three data sets, pairwise, at different p-value network thresholds and gamma parameters for community detection

Shown is 1-normalised variation of information as throughout.

Click here for additional data file.

Table S6 Table summarising the OTUs within each of the community types identified across all three datasets

Click here for additional data file.

Additional Information and Declarations

Competing Interests

Author Contributions

Human Ethics

Data Availability

Tim D. Spector is co-founder of MapMySelf and MapMyGut Ltd. All other authors declare that they have no competing interests.

Matthew A. Jackson conceived and designed the experiments, performed the experiments, analyzed the data, contributed reagents/materials/analysis tools, wrote the paper, prepared figures and/or tables, reviewed drafts of the paper.

Marc Jan Bonder performed the experiments, analyzed the data, contributed reagents/materials/analysis tools, reviewed drafts of the paper.

Zhana Kuncheva conceived and designed the experiments, contributed reagents/materials/analysis tools, reviewed drafts of the paper.

Jonas Zierer reviewed drafts of the paper.

Jingyuan Fu, Alexander Kurilshikov, Cisca Wijmenga, Alexandra Zhernakova, Jordana T. Bell, Tim D. Spector and Claire J. Steves contributed reagents/materials/analysis tools, reviewed drafts of the paper.

The following information was supplied relating to ethical approvals (i.e., approving body and any reference numbers):

Ethics approval for the TwinsUK study was given by the NRES Committee London—Westminster (REC Reference No.: EC04/015), with all participants providing written consent. The Lifelines-DEEP study was approved by the institutional ethics review boards of the University Medical Centre Groningen (ref. M12.113965).

TwinsUK 16S rRNA gene sequencing data is available from the ENA (Accession: PRJEB13747). LLDEEP 16S rRNA gene sequencing data is available from the EGA upon request of an account from the Lifelines-DEEP group (Accession: EGAD00001001991). The data from the Israeli-PN study was obtained from the ENA (Accession: PRJEB11532). Code used for the main stages of OTU and co-occurrence network creation and the identification of communities within the networks can be found in the Supplementary Materials.

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
