# Peer review of "Detection of stable community structures within gut microbiota co-occurrence networks from different human populations"

_PeerJ, doi:10.7717/peerj.4303_

## Round 0.1 · original submission · Major Revisions

Your paper has been reviewed by three experts in the field, and while two reviews were very positive, requiring only minor revisions, the third reviewer has apart from minor revisions and suggestions also two serious concerns with respect to the validity of your findings, one regarding the robustness of the qualitative conclusions, and one regarding detected communities that could be potentially artifacts.

Both aspects need to clarified before the paper can be accepted. Please adress in your revision not only those main concerns but also the minor revisions or suggestions by all three reviewers either by following their advice or by giving a solid argument why not to do so in your Point-to-point response.

Reviewer 1 ·

Basic reporting

The language is ok, but some mistakes should be fixed (see General comments).

Literature references & relevance: reference format should be fixed in a couple of places in the manuscript.

Raw data: «LLDEEP 16S rRNA gene sequencing data is available from the EGA (Accession: EGAD00001001991).» - However, at the specific location, the reads are in restricted access.

Experimental design

Identification of co-occurence groups in microbiota is a topic of high interest. Advance in this area will make metagenomic data analysis and interpretation more robust and insightful. The manuscript contributes to the field via large-scale meta-analysis performed using a novel verified ensemble-based methodology.

The methods are described in details.

Validity of the findings

The findings are well supported.

Additional comments

Intro & background: As far a reviewer is concerned, in order to make the introduction more rounded, reference to co-occurence modeling in relation to metabolic interactions should be added (http://www.pnas.org/content/110/31/12804.full ) - as well as to the algorithms for inference of microbial co-occurrence from temporal points analysis (not just single metagenomes).

Could you comment on how would the main outcomes of the study change if the OTU are generate not at 97% but at higher or lower percent of identity?

There is another method for co-occurrence network analysis - SPIECEASI - that is related to SparCC but reported to be superior to it [http://journals.plos.org/ploscompbiol/article?id=10.1371/journal.pcbi.1004226]. Addition of it to the set of analyzed algorithms would add value to the manuscript, as far as the reviewer is concerned.

Other specific comments - line-wise

Line #23: "The interactions defining these communities can be inferred from the co-occurrence of taxa across multiple samples" - not always: the microbes might just share 1 niche and not interact at all (e.g. see the conclusions of http://www.pnas.org/content/110/31/12804.full)

38 - "in either an actively targeted or passive manner" - it can be neutral, too

50 — specialist approaches - specialized approaches might sound better.

57 - subsets OF taxa

86 - It is worth noting that that the sequencing platform was the same in all studies.

109 - "de novo" - should be in italic here and elsewhere. Same applies to "in vitro".

131 - Is there any rationale behind the number 25%?

(many times in the text) Perhaps, "chosen" should be replaced with "selected".

155 - misprint: compossitionality

161 - unfinished title

166 - Why do authors use different adjustment method (Bonferroni - while applying FDR in other analyses)?

176 - From biological point of view, it was not completely clear to the reviewer why was the fitting of network structure to scale-free network used to identify the optimal p-value. Perhaps, this could be clarified.

177 - "whose" - please replace here and elsewhere:
We use "who" when referring to people or when we want to know the person.
We use "which" to refer to a thing or an idea, and to ask about choices.
We use "that" for both a person and a thing/idea.

182 - R2 - 2 should be superscript

186 - used -> was used

210 - sub set - ?

Fig 2B - Could you comment on the impulses observed for Israeli data in top 2 plots? Looks like potential inconsistency in the data - in comparison with the overall smooth plots.

340 - please explain the meaning of the "mixture of assignments" - here or in Methods. Also, in Suppl. Table 1 it is not clear which of the assignments are "mixture".

353 - "heritable taxa" - the term appears to be unclear. Strictly, the heritability of gut microbes has not been demonstrated - we can only talk about its inferred extent. Consider - "highly heritable taxa".

357 - associated -> were associated

423 - With BMI, the direction of causality is not that certain.

436 - Consider citing some of the recent high-impact articles from 2017 about this topic (for instance, https://www.nature.com/articles/nbt.3981 or https://www.ncbi.nlm.nih.gov/pubmed/28967887).

476, 497 - reference format.

488 - butryate

489-492 - The logic is not clear. Moreover, the 2006 study is about mice and the results have not been confirmed in humans in a large-scale meta—analysis (http://mbio.asm.org/content/7/4/e01018-16.full).

492 - Bifidobacterium - should be in italic.

493 - As far as reviewer is concerned, Yatsunenko et al have reported the descrease of Bifidobacterium for other age cohorts - prevailed by infants. Might be better to rephrase.

494 - negatively -> were negatively

·

Basic reporting

The article is generally very well written. The abstract and introduction suffer from typographic errors - for example line 30 is missing an adverb (‘significantly more similar’), line 57 a preposition (‘subsets of taxa’). A simple proofread will correct this.

Weiss et al. 2016 is a joint first author paper. Perhaps it should read Weiss and Van Treuren et al. 2016 here and in other locations where it is cited.

Experimental design

From a methodological perspective, the paper is well thought out. The explicit description of the algorithmic steps taken to generate final modular network make it clear the authors evaluated each choice in this pipeline. The description and exploration of the important parameters (e.g. gamma) is admirable and more papers would do well to emulate this. Certain choices seem somewhat arbitrary, but that is inevitable in a pipeline that transforms the data through multiple steps.

Validity of the findings

The findings are valid and the data accessible.

Additional comments

Jackson et al. present a method for detecting/building taxa co-occurrence graphs and then analyze a co-occurrence network built from three large human 16S sequencing datasets. Primary network building is accomplished by selecting the intersecting edges of graphs generated with four independent detection techniques following recommendations of Weiss and Van Treuren et al. The stringency of edge detection (p-value) is set by choosing the p-value that best produces a scale-free network. Finally, communities are detected in the network by aggregating the edges (pairwise co-occurrences) into modules based on local connection and modularity.
To validate this method of community detection the authors recapitulate findings from Goodrich et al. 2014 in the same dataset used in that paper. They also find associations between host subject parameters (e.g. BMI) and community abundance that are supported by recent literature. To conclude, the authors suggest their method creates analytical tractability and dimensionality reduction while capturing the important ecological units that are operating across population in the human gut.
From a methodological perspective, the paper is well thought out. The explicit description of the algorithmic steps taken to generate final modular network make it clear the authors evaluated each choice in this pipeline. The description and exploration of the important parameters (e.g. gamma) is admirable and more papers would do well to emulate this. Certain choices seem somewhat arbitrary, but that is inevitable in a pipeline that transforms the data through multiple steps. While the methodology is clearly described, the benefit of the final network in data analysis, hypothesis generation, experimental motivation, etc. is unclear. The authors could strengthen the manuscript by indicating what their network approach enables that was unattainable with previous methods.

Reviewer 3 ·

Basic reporting

The manuscript is clearly written, and provides a good background and literature references. I felt like a lot of the technical details regarding the analysis (e.g. choice of gamma parameter) disrupted the flow of the manuscript. I would significantly shorten these parts of the "Results" section and defer the more detailed description to the "Materials and Methods"and supplement.

Additionally, the use of the word "community" may be confusing to some readers. It is used in this work as a technical term from network science, representing groups that are more likely to be linked (correlated) within a group than between groups. However, for microbiome researchers and ecologists the term is used more generally to describe the full set of taxa that reside in the gut. This is a subtle difference that may be worth clarifying explicitly.

Experimental design

This is an original primary research with well defined questions addressing a current gap in our knowledge. Specifically, the authors ask whether similar groups of correlated microbial taxa are found in the intestinal tracks of distinct groups of people, and whether such groups show similar correlations with host properties.

Sufficient detail are given for these results to be replicated. I would just add references to the QIIME and VSEARCH publications.

Validity of the findings

Overall, this is a well executed study, and the authors do a good job highlighting potential caveats. Nonetheless, I have two main concerns regarding the validity of the conclusions.

First, the authors have made several somewhat arbitrary choices in the analysis. These include the choice of the ensemble methods for inferring correlations, fit to a scale-free networks for determining p-values threshold, procedure for picking gamma, use of unweighted networks that don't account for the magnitude of correlations, and the use of different multiple-hypothesis corrections for different datasets. While these are mostly reasonable choices, the results may be sensitive to some of these choices. It would substantially strengthen the validity of the conclusions if they can be shown to be robust to variations in these arbitrary choices. I realize that changing these details would affect the specifics of the networks and detected communities, but the robustness of the qualitative conclusions should be assessed.

Second, I am concerned that many of the detected communities do not represent biologically meaningful groups of correlated taxa, but rather artifacts associated with sequencing errors and chimeras. The authors could try and control for such effects by grouping OTUs with a lower identity threshold, using closed-reference OTU calling, and focusing on communities composed of phylogenetically-diverse taxa.

---

## Round 0.2 · Minor Revisions

Thanks for having done a great job in revising the manuscript. As you can see from the reviewers comments all major concerns have been addressed properly and only a few minor, but still important points with respect to the sensitivity of the results to the chosen gamma threshold and the p-value threshold still need more clarification. Please check the detailed remarks of reviewer three. I agree with the reviewer that showing that the results are more robust to some of the thresholds would improve the manuscript by strengthening the findings and improving the confidence in the presented work.

Reviewer 1 ·

Basic reporting

Please remove extra "the" at the beginning of the abstract.

Line 560 - Bifidobacterium - should be in italic.

Experimental design

No comments

Validity of the findings

No comments

Additional comments

All remarks have been addressed thoroughly.

Reviewer 3 ·

Basic reporting

No comment.

Experimental design

No comment.

Validity of the findings

I thank the authors for their detailed response, adding details, clarifications, and novel analysis that has mostly addressed my main concerns. Nonetheless, I still feel there are a few places where showing that the results are more robust to some of the thresholds would strengthen the findings. In addition, providing more details about the choice of thresholds (as detailed below) would help others implement this analysis on additional datasets.

Specifically, I appreciate the authors' points regarding the utility of picking a p-value threshold based on the fit to a scale-free network. However, it still seems warranted to check the sensitivity of results to the selected p-value, as should be done for any other method for picking thresholds. For example, I would be concerned if analysis based on a different p-value threshold, say 0.005, that fits a bit worse to a scale-free network results in very different communities that are not conserved across datasets.

In addition, in these datasets, it was fortunate that a single p-value threshold offered good fit for all datasets, and resulted in approximately the right slope (This seems very fortunate indeed, given the erratic behavior of the fit for the Israeli-PN dataset). But what should be done if such a threshold does not exist? Should a different threshold be chosen for each dataset? What if several thresholds fit? What should be done if one threshold offers a better R2, while another a slope closer to the desired one?

Similarly, there is no unique, obvious choice of gamma threshold. The authors very reasonably pick a threshold that results in reproducible communities and statistically significant modularity. However, looking at Figure 3, other values may serve as well, say 0.2. I would be more confident in the results if the authors checked how sensitive are results to this choice, and elaborate on how exactly it is made in this case and in general (as above, when no single threshold fits all datasets, and there are trade-offs between reproducibility and significance within datasets).

---

## Round 0.3 · accepted · Accept

All remaining concerns regarding the robustness of the results to some of the thresholds and details about the choice of thresholds have been addressed to my satisfaction in this revised manuscript. would, strengthen the findings. In addition, I feel that the revision made the manuscript really stronger and more convincing, good job!